



# The long-term spatio-temporal variability of sea surface temperature in the Northwest Pacific and the Near China Sea

**Zhiyuan Wu [1,2,3], Changbo Jiang [1,3,*], Mack Conde [4], Jie Chen [1,3], Bin Deng [1,3]**

[1] School of Hydraulic Engineering, Changsha University of Science & Technology, Changsha, 410114, China

[2] School for Marine Science and Technology, University of Massachusetts Dartmouth, New Bedford, MA 02744, USA

[3] Key Laboratory of Water-Sediment Sciences and Water Disaster Prevention of Hunan Province, Changsha, 410114, China

[4] School of Marine Science and Ocean Engineering, University of New Hampshire, Durham, NH 03824, USA

\* Correspondence: chbjiang@csust.edu.cn

**Abstract:** The variability of the sea surface temperature (SST) in the Northwest Pacific has been studied on seasonal, annual and interannual scales based on the monthly datasets of ERSST 3b (1854-2017, 164 years) and OISST V2 (1988-2017, 30 years). The overall trends, spatial-temporal distribution characteristics, regional differences in seasonal trends, and seasonal differences of SST in the Northwest Pacific have been calculated over the past 164 years based on these datasets. In the past 164 years, the SST in the Northwest Pacific has been increasing linearly year by year with a trend of 0.033 °C/10 yr. The period from 1880 to 1910 is a slow decreasing trend period in the past 164 years and the SST during the 1910-1930 period was a trough of the past 164 years. After 1930, SST has continued to increase until now. The increasing trend in the past 30 years has reached 0.132 °C/10 yr and the increasing trend in the past 10 years is 0.306 °C/10 yr, which is around ten times in the past 164 years. The SST in most regions of the Northwest Pacific showed a linear increasing trend year by year, and the increasing trend in the offshore region was stronger than that in the ocean and deep-sea region. The change trend of the SST in the Northwest Pacific shows a large seasonal difference, and the increasing trend in autumn and winter is larger than that in spring and summer. There are some correlations between the SST and some climate indexes and atmospheric parameters, the correlation between the SST and some atmospheric parameters have been discussed, such as NAO, PDO, SOI anomaly, TCW, Nino 3.4, SLP, Precipitation, T2 and wind speed. The lowest SST in the Near China Sea basically occurred in February and the highest in August. The SST fluctuation in the Bohai Sea and Yellow Sea (BYS) is the largest with a range from 5 °C to 22 °C, the SST in the East China Sea (ECS) is from 18 °C to 27 °C, the smallest fluctuations occurs in the South China Sea (SCS) maintained at range of 26 °C to 29 °C. There are large differences between the mean and standard deviation in different sea regions.

**Keywords:** sea surface temperature; spatio-temporal distribution; interannual and interdecadal time scales; the Northwest Pacific



**1. Introduction**


The ocean is one of the important components of the ocean-atmosphere coupling system (Chelton
and Xie, 2010; Wu et al., 2019a,b). Relative to the atmosphere, the ocean has characteristics such as slow
change and large heat capacity (England et al., 2014). Because of the gradual changes in the ocean, climate
change at the interannual, decadal, and longer timescales may be closely related to the ocean (Trenberth
and Hurrell, 1994; Ault et al., 2009). The Sea Surface Temperature (SST) is the basis for the interaction
between the ocean and the atmosphere (Wu et al., 2019c,d), and it characterizes the combined results of
ocean heat (Buckley et al., 2014; Griffies et al., 2015), dynamic processes (Takakura et al., 2018). It is a
very important parameter for climate change and ocean dynamics process, reflects sea-air heat and water
vapor exchange. Observations and numerical simulations show that large-scale sea surface temperature
anomalies of over 20° in longitude and latitude can cause significant changes in atmospheric circulation,
such as the El Niño and La Niña phenomena (Chen et al., 2016; Zheng et al., 2016). During El Nino, the
trade winds in the tropical East Pacific will be weakened, and the SST increased significantly, which was
3~5°C higher than normal years.   As a result, major changes have been made in the atmospheric
circulation and ocean circulation, which has caused the worldwide atmospheric and marine environment
and the abnormality of climate (Li et al., 2017).

The Northwest Pacific is particularly affected by the El Niño in the East Pacific and determines the
oceanic climate change in China (Hu et al., 2018). On one hand, climate change causes an increasing SST
in the northwestern Pacific, which increases the vertical stratification of the water, affects the atmospheric
circulation, and changes the intensity and period of coastal winds and upwelling. On the other hand, the
10-year periods Pacific Decadal Oscillation (PDO) and the El Niño-Southern Oscillation (ENSO) occur
on average every 2 to 7 years, resulting in large variations in upwelling (Xiao et al., 2015; Yang et al.,
2017; Xue et al., 2018). These factors will all lead to the impact on the marine environment in Chinese
coastal areas, causing land-based droughts and floods and climate disasters (Xu et al., 2018). Therefore, it
is very urgent to study the impact of climate change on SST in the Northwest Pacific and the Near China
Sea. As one of the main parameters of global climate change and one of the important characterizations
and predictors of El Niño, the study of SST changes is particularly important.

Previous scholars have done a lot of work on the changing trend of SST. According to the Fifth
Assessment Report (AR5) of the Intergovernmental Panel on Climate Change (IPCC), the global SST
warming trend was 0.064 °C/10 yr between 1880 and 2012 (Pachauri et al, 2014). In fact, many studies
have shown that the Pacific SST anomalous changes are closely related to global and regional climate
changes, and it has multi-scale temporal variations (Graham, 1994; Latif, 2006; Shakun and Shaman, 2009;
Li et al, 2014). In addition, the El Niño-Southern Oscillation (ENSO) and the Pacific Decadal Oscillation
(PDO), which are closely linked to global and regional climate change, are found in this area. Therefore,
the Pacific is one of the key ocean areas that scholars have studied for a long time (Bao and Ren, 2014;
Mei et al., 2015; Stuecker et al, 2015; Wills et al, 2018).

So far, two types of main meteorological SST datasets have been obtained: one based on measured
mid-resolution (1° -5°) 100-year datasets and the other based on satellite high-resolution (1-10km) decade



datasets (Wang et al., 2011; Smith et al., 2014; Huang et al., 2015, 2016; Diamond et al., 2015). The former
has rebuilt a time series of months over 150 years and the latter has accumulated over 30 years of time
series on a daily average basis (Tian et al., 2019). The existing climatic datasets already have conditions
for allowing the creation of a natural mode of change in SST in terms of duration and resolution (Liu et
al., 2017; Wang et al., 2018). With the continuous improvement of ocean observation technology and the
accumulation of satellite remote sensing data, the conditions for the scholars use the satellite data for short-
term climate change research have been met. In recent years, the research and discussion on the interannual
change of SST based on satellite remote sensing SST has attracted wide attention (Tang et al., 2003; Yang
et al., 2013; Zhang et al., 2015; Skirving et al., 2018).

Satellite remote sensing can achieve large-area simultaneous measurements with high temporal and

spatial resolution. The remote sensing SST obtained is conducive to a more comprehensive and rapid
understanding of oceanographic phenomena that affect the ocean surface, including El Niño (Robinson,
2016). At present, about 30 years of satellite remote sensing SST data have been accumulated (Franch et
al., 2017), and a set of sea surface temperature data has been provided to study the conditions for the
occurrence and development of ocean surface heat change modes in the temporal and spatial span and
resolution. So, satellite remote sensing SST has received widespread attention in recent years.

At present, based on satellite remote sensing data, the time scales for the study of changes in SST in

the Northwest Pacific, especially in the Near China Sea, are mostly within 20 years, which is relatively
short for studying climate change (Song et al., 2018; Pan et al., 2018). Most of the space is targeted at
specific local sea areas, and there is less research on the changes of the SST in the Northwest Pacific
covering all marginal seas of China. Therefore, it is necessary to study the SST variation of large-scale
and long-term sequences based on satellite remote sensing data.

Previous scholars have made great contributions to the study of global warming, but most of them

are the overall changes in the regional average SST, and they tend to ignore the characteristics of changes
in certain key sea areas. There are great differences in the trends of SST in different sea areas. The long-
term trend of the SST changes in the Northwest Pacific (0° N- 60° N, 100° E- 180° E) over the past 164
years (1854-2017) have been calculated based on the monthly datasets of ERSST 3b in this study. The
temporal and spatial distribution characteristics of SST, the overall long-term sequence variation trend,
the regional variation of the seasonal trend, and the seasonal differences were analyzed. The correlations
with SST changes and climate parameters and indexes are been analyzed. To provide a reference for the
study of global climate change, the characteristics of SST changes in the Near China Sea has been studied
in this paper.

High spatial resolution SST datasets including average SST field and monthly SSTA field are been

obtained. In view of the fact that there are many interannual and intra-annual changes, this paper analyzes
the characteristics of SST changes based on these datasets. The trend, inter-decadal changes in SST and
their causes, and the correlation with the climate parameters and indexes such as Nino-3.4 index are
relatively low. The ocean thermal dynamic phenomenon is preliminarilly discussed. The datasets are
processed and analyzed to study the trend changes of the SST in the Northwest Pacific. To explore the



correlation and response mechanisms with climate systems such as the ENSO and the PDO, and to conduct

a detailed analysis of typical sea areas.

**2. Study region, Data and Methods**

*2.1. Study Region*

The Northwest Pacific is the northwest region of the Pacific, are defined as the offshore region of

0°N- 60°N and 100°E - 180°E in this study (Fig.1). There are more tropical cyclones over the Northwest

Pacific than any other sea area in the world, with an average annual average of 35. About 80% of these

tropical cyclones will develop into typhoons. On average, about 26 tropical cyclones per year reach at least

the intensity of tropical storms, accounting for about 31% of the global tropical storms, and more than

double the number of any other area. The sea-air interaction in this area is very strong and the change of

SST is worth to explore.

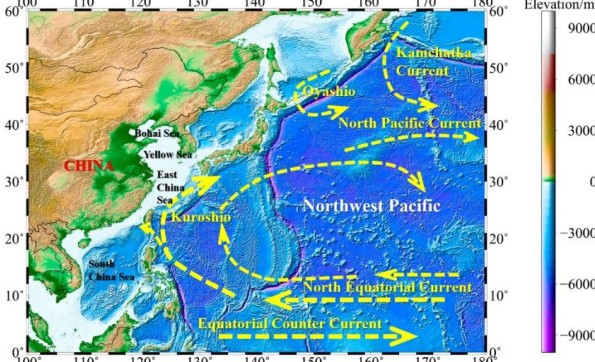

**Figure 1.** Bathymetric map of the Northwest Pacific and ocean circulation.

*2.2. SST Dataset*

Several data sources are used to analyze the long-term temporal and spatial variability of SST in the

Northwest Pacific in this present study. Long-term statistics are based on the monthly SST data from the

Extended Reconstructed Sea Surface Temperature (ERSST) 3b (1854-2017) (Smith et al., 2008). The

ERSST dataset is a global monthly sea surface temperature analysis derived from the International

Comprehensive Ocean–Atmosphere Dataset with missing data filled in by statistical methods. This

monthly analysis begins in January 1854 continuing to the present (https://www1.ncdc.noaa.gov/

pub/data/cmb/ersst/v3b/). The primary SST dataset analyzed in this study is the NOAA Optimum

Interpolation (OI) Sea Surface Temperature (SST) V2 (OISST V2 1982-2017, http://www.esrl.noaa.gov/

psd/data/gridded/data.noaa.oisst. v2.html) (Reynolds et al., 2002, 2007). The advantage of this dataset is

apparent when compared with other gridded datasets such as HadISST, ERSST and OSTIA, which spans

only the period since 2007.

The seasonal mean data are obtained by averaging the monthly average SST after the above-

mentioned processing. The spring is March, April and May (MAM), the summer is June, July and August



(JJA), the autumn is September, October and November (SON), and the winter is December of the
previous year and January and February (DJF).
The SST anomaly is the deviation from the long-term SST average of the observations of the SST
describing a particular area and time. The year anomaly represents the deviation of the average of the SST
for a given year from the mean of the multi-year SST. The month anomaly represents the deviation of the
average of the SST for a particular month from the average of the SST for that particular month for many
years. In this paper, the mean value from 1854 to 2017 is taken as the climate mean state, and the sea
surface temperature anomaly is subtracted from the SST field to obtain the SSTA field.
*2.3. Climate Index Dataset*
The Atlantic Multidecadal Oscillation (AMO) is a climate cycle that affects the sea surface
temperature (SST) of the North Atlantic Ocean based on different modes on multidecadal timescales
(http://www.esrl.noaa.gov/psd/data/timeseries/AMO, McCarthy et al, 2015). Niño 3.4 index uses SST to
characterize ENSO, the Niño 3.4 SST region consists of temperature measurements from between 5° N -
5° S and 120° - 170° W (Gergis and Fowler, 2005).The PDO index is the time coefficient of the first mode
obtained by performing EOF of the mean SSTA in the north of 20° N in the North Pacific
(http://jisao.washington.edu/pdo/PDO.latest). The North Atlantic Oscillation (NAO) is the most
prominent modality in the North Atlantic. Its climate impact is most prominent mainly in North America
and Europe, but it may also have an impact on the climate in other regions such as Asia. Recent studies
have not only further confirmed its existence, but also revealed its connection with a wide range of oceans
and atmospheric conditions.
The correlation between the SST and the atmospheric parameters is analyzed based on the ERA-
Interim data. ERA-Interim refers to the European Centre for Medium-Range Weather Forecasts (ECMWF),
which is an independent intergovernmental organization supported by 34 countries. Its goal is to develop
numerical methods for mesoscale weather forecasting. The country provides forecasting services, conducts
scientific and technological research to accumulate forecasts, and accumulates meteorological data. ERA-
Interim is the latest global reanalysis product developed by ECMWF. The weather data and climate data
from January 1988 to December 2017 are used in this paper, such as sea surface temperature, sea-to-air
interface heat flux, and wind field data at a height of 10m, the spatial resolution of these datasets is
1.5°×1.5°.
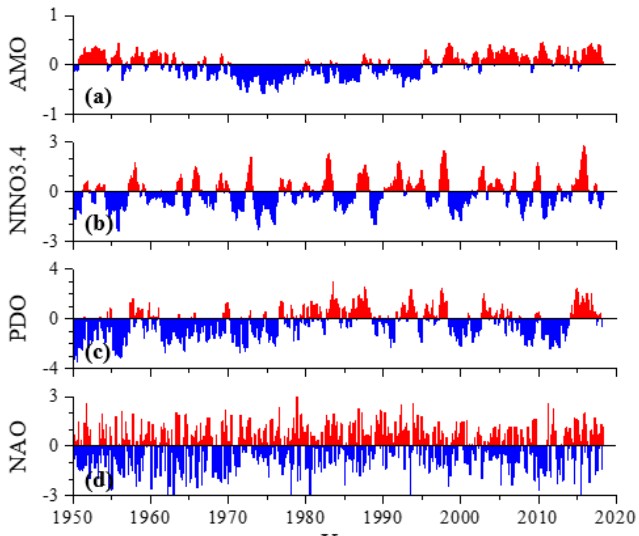

**Figure 2.** AMO index (a), Niño 3.4 index (b), PDO index (c) and NAO index (d) during
1950~2017.

*2.4. Methods*

Regression analysis is an important part of mathematical statistics and multivariate statistics. It is a mathematical method to study the correlation between variables and variables. The regression analysis has a wide range of applications in the statistical forecasting of oceans and atmospheres. It is used to analyze the statistical relationship between a variable (called forecast) and one or more independent variables (called predict), and to establish a forecast. The regression equation produced by the quantity and forecast factor, and then based on this equation to make predictions of the forecast volume. Regression analysis includes linear regression and nonlinear regression. The linear regression is commonly used, and a linear regression analysis method is used in this paper.

Use $x_i$ to represent a climate variable with a sample size of $n$. Use $t_i$ to represent the time corresponding to $x_i$ and establish a linear regression between $x_i$ and $t_i$. The formula can be expressed as:

$$x_i = a + bt_i, \quad i = 1, \ 2, \ 3, \ ..., \ n \tag{1}$$

Where, $a$ is the regression constant and $b$ is the regression coefficient. $a$ and $b$ can be calculated using the least squares method.

For the observation data $x_i$ and the corresponding time $t_i$, the least-squares calculation result of the regression coefficient $b$ and the constant $a$ is expressed as:

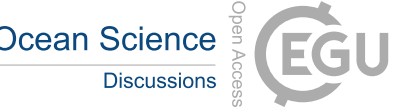
$$b = \frac{\sum_{i=1}^{n}\left(x_i - \bar{x}\right)\left(t_i - \bar{t}\right)}{\sum_{i=1}^{n}\left(x_i - \bar{x}\right)^2} \tag{2}$$

$$a = \bar{x} - b\bar{t}$$

The correlation coefficient between time $t_i$ and $x_i$ is:

$$r = \sqrt{\frac{\sum_{i=1}^{n} t_i^2 - \frac{1}{n}\left(\sum_{i=1}^{n} t_i\right)^2}{\sum_{i=1}^{n} x_i^2 - \frac{1}{n}\left(\sum_{i=1}^{n} x_i\right)^2}} \tag{3}$$

The correlation coefficient $r$ is expressed as the degree of closeness of the linear correlation between
the variable $x$ and the time $t$. When $r > 0$, $b > 0$, indicating that $x$ increases with time $t$; when $r < 0$, $b < 0$,
indicating that the variable $x$ decreases with time $t$. Perform a significant test on the correlation coefficient
to determine the significance level $\alpha$ (confidence is 1-$\alpha$) first. If $|r| > r_\alpha$, shows that the trend of the
variable $x$ with time $t$ is significant, otherwise it is not significant.
**3. Results and Discusses**
*3.1. Temporal distribution of SST*
With the gradual warming of the global climate, the average temperature of the ocean is also rising.
In order to reflect the overall trend of SST in the Northwest Pacific over the past 164 years (1854-2017),
the average monthly SST data from 1854 to 2017 was used. The time series curve of SST in the Northwest
Pacific, the Northern Hemisphere, and the global ocean was obtained by processing, and the overall trend
of the SST was analyzed, as shown in Fig. 3. As can be seen from the figure, SST in the different region
have shown an increasing trend and SST has shown a significant increasing trend since the 20th century.
The SST datasets were used to calculate the SST anomaly time series and its linear variation trend in
the Northwest Pacific, the Northern Hemisphere and the global ocean as shown in Fig. 3. The slope of the
linear equation with one unknown obtained by least-squares fitting is the annual change rate of SST, as
shown in Table 1. It shows the increasing trend of SST at different time scales. It can be seen that the data
shows that the SST in the different region has shown a significant warming trend as a whole. It can be
seen from Table 1 that from 1854 to 2017, the SST trend of Northwest Pacific, North Hemisphere and
global ocean has increased by 0.033 °C to 0.035 °C per 10 years. In the past 50 years, the increasing rate
of SST has reached 0.10 °C/10 yr or more, and the increasing rate in the last 10 years has reached 0.30°C.
It can be seen that the warming trend of SST in the Northwest Pacific is very significant.





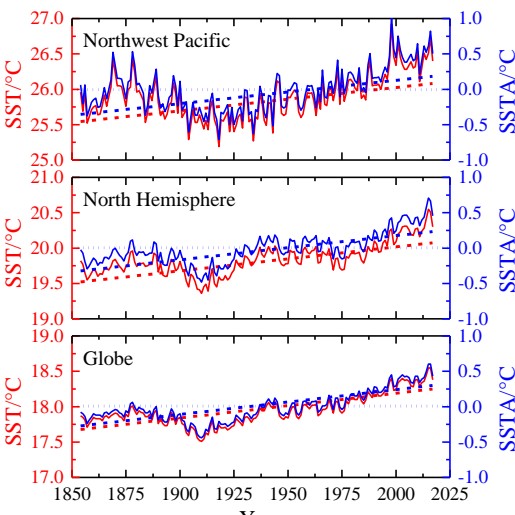

**Figure 3.** The temporal variability of annual SST.

**Table 1.** The average trend of SST (Unit: °C/10 yr).

|  | NWP | NH | GLO |
|---|---|---|---|
| 1854-2017 (164yr) | 0.033 | 0.034 | 0.035 |
| 1918-2017 (100yr) | 0.100 | 0.059 | 0.069 |
| 1968-2017 (50yr) | 0.128 | 0.128 | 0.102 |
| 1988-2017 (30yr) | 0.132 | 0.149 | 0.102 |
| 2008-2017 (10yr) | 0.306 | 0.379 | 0.274 |

NWP: Northwest Pacific; NH: North Hemisphere; GLO: Globe. All the trend is significant at the 95% confidence level.

There exist decadal to multi-decadal variations in the SST and SST anomalies series, with a general cool period from the 1880s to 1910s, a weak warm period from 1920s to 1940s, a weak cool period from 1970s to 1980s, and a recent warm period from 1990s to present. Fig.3 also show that the interannual to decadal variability is larger in the Western Pacific, and it is smaller in the global ocean, indicating an increase in SST anomaly variability with the area. It is also interesting to note that the latest 10 years see a larger increasing trend of annual mean SST than that for the last 164 years, 100 years, 50 years and 30 years, indicating an obvious speed-up of warming of the Northwest Pacific, North Hemisphere and globe ocean occurs in the last 10 years, and the growth rate over the past decade has been around ten times that of the past 164 years.

In the past 164 years, the correlation coefficient of SST trends in the Northwest Pacific was 0.73. It passed the 95% reliability test, which shows that the linear trend is significant, and the regression coefficient is 0.0033. This shows that in the past 164 years, the SST in the Northwest Pacific has been increasing linearly year by year at a rate of 0.033 °C/10 yr. It can be seen from Fig. 3 that during the period of 1870-1910, it showed a slowly decreasing trend, SST basically fluctuates slightly between 25.2 °C to

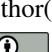



26.0 °C; during the period of 1910-1930, the SST is the valley of nearly 164 years, and the curve trend is
also very gentle; after 1930, the SST oscillated gradually, and the trend has continued to this day.
In order to demonstrate the seasonal variation of the SST trend in the Northwest Pacific, the SST at
1°×1° at each grid point in the Northwest Pacific was averaged from 1854 to 2017 by winter, spring,
summer, autumn and year in this study. The season-by-season linear trend of SST at each grid point has
been analyzed. At the same time, the season-by-season time series of the SST anomalies were being
calculated and the seasonal variation of the comparison trends was shown in Fig 4.
Fig.4 (a) and (b) show seasonal and annual mean SST and SST anomalies series. The blue lines are
their trends of every seasonal mean SST and SST anomalies series for the Western Pacific during 1854-
2017, the red lines is their trends during 1988-2017. The increasing trends during 1854-2017 is between
0.032 °C/10 yr and 0.035 °C/10 yr for all seasons. The same as the annual pattern, seasonal pattern for the
latest 30 years shows more significant warming trend than 164 years. Significant warming occurs in all
seasons with those of autumn and winter being the largest, reaching 0.146 °C/10 yr and 0.124 °C/10 yr
respectively at the last 30 years, and that of spring the smallest.
The magenta points mean the SST anomaly larger than 0.4 °C, and the cyan points mean the SST
anomaly smaller than -0.4 °C in the Fig.4 (b). As can be seen from the figure, during the period from 1890
to 1960, there were more negative anomalies and less than -0.4 °C, indicating that there was a cool period
during this period. In the period from 1988 to 2017, there are more positive anomalies and more than
0.4 °C, indicating that there is a warm period in the past 30 years.
In the analysis of the SST changes in the Northwest Pacific during the past 164 years, it has been
found that there was a strong warming trend in SST over the past 30 years since 1988. It had been shown
that the SST in the Northwest Pacific has an overall warming trend since the 1970s in the previous studies
(Zhou et al., 2009; Kosaka et al., 2013) and this study. The time series curve of the SST in the Northwest
Pacific from 1988 to 2017 was plotted as shown in Fig. 4(c).
Yamamoto's (1986) method has been used to determine the mutation point, and the formula is:

$$R_{SN} = \frac{\left|\overline{X_1} - \overline{X_2}\right|}{S_1 + S_2} \tag{4}$$

Where, $\overline{X_1}$, $\overline{X_2}$, $S_1$, $S_2$ are the average and standard deviation of the two stages before and after
the mutation year. It was found that there were six stations when $X_1 = X_2 = 10$, $R_{SN} \geq 0.7$ in 10 years before
and after 1998/1999, and the significance level of the statistic reached $\alpha = 0.05$, according to which the
SST was considered to have a mutation in this year. The difference between the mean value of the anomaly
before and after the mutation was 0.30°C, and the similar results can also be seen in Fig. 4(c). It can be
found that in the past 30 years, the SST in the Northwest Pacific has significantly warmed up as a whole.
The highest annual mean SST appears in 1998, and the temperature undergoes a weak decreasing trend
since then, but the average SST during 1998-2007 reaches 26.446 °C, which is higher than around 0.3 °C
during 1988-1997. In the last 30 years of SST in the Northwest Pacific, the increasing trend in the last 10
years is obviously greater than the trend in the last 30 years.

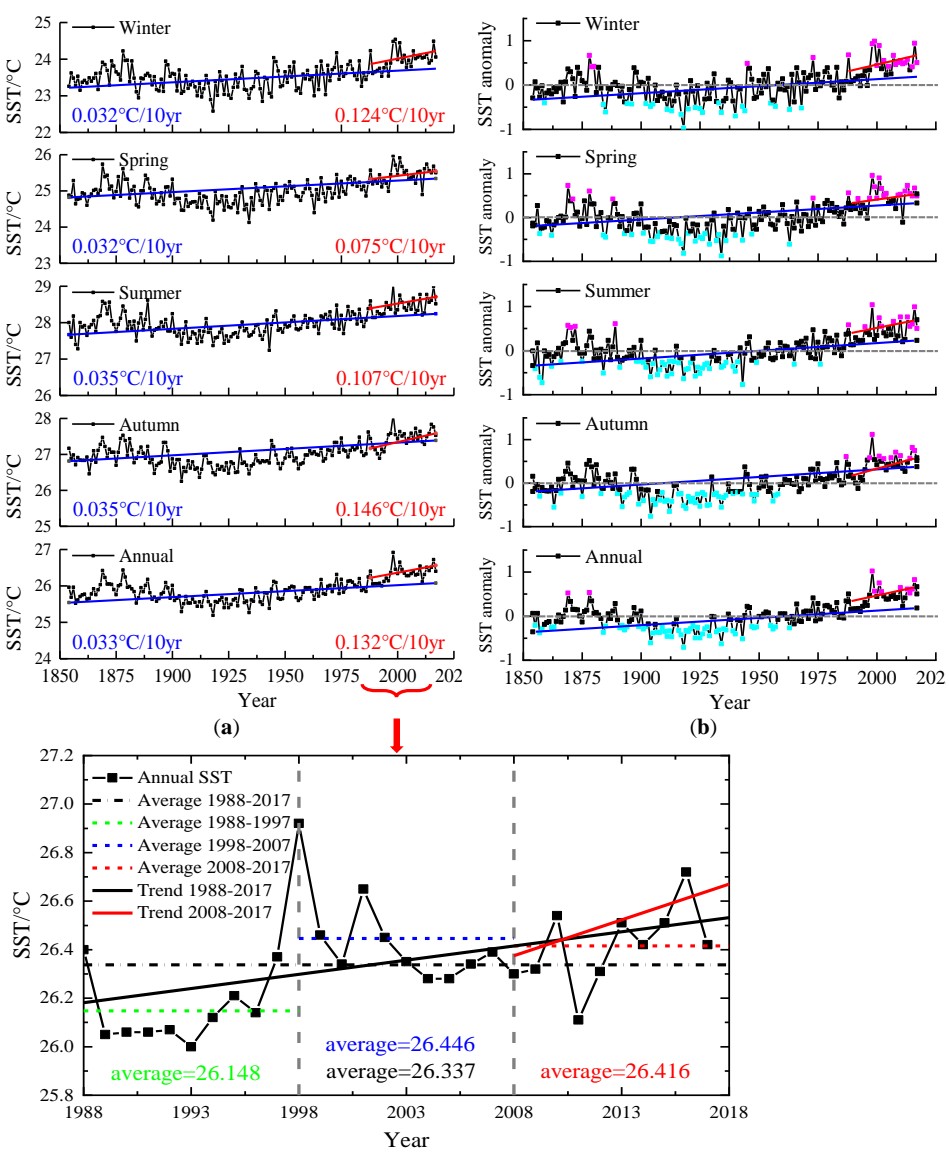

**Figure 4.** Variability of seasonal/annual SST. (a) the annual SST over the 1854-2017 period; (b) the SST anomaly over the 1854-2017 period; (c) the SST over the 1988-2017 period (the latest 30 years).

The monthly average sea surface temperature in the Northwest Pacific is represented by an undulating curve, as shown in the blue dashed line in Fig. 5, and the sea surface temperature anomaly is a red dotted line. The positive value is filled in yellow, and the negative value is filled in cyan. The NINO3.4 index is one of several El Niño/Southern Oscillation (ENSO) indicators based on sea surface temperatures. NINO3.4 is the average sea surface temperature anomaly in the region bounded by 5°N to 5°S, from



170°W to 120°W. This region has large variability on El Niño time scales, and is close to the region where
changes in local sea surface temperature are important for shifting the large region of rainfall typically
located in the far western Pacific. An El Niño or La Niña event is identified if the 5-month running-average
of the NINO3.4 index exceeds +0.4°C for El Niño or -0.4°C for La Niña for at least 6 consecutive months.

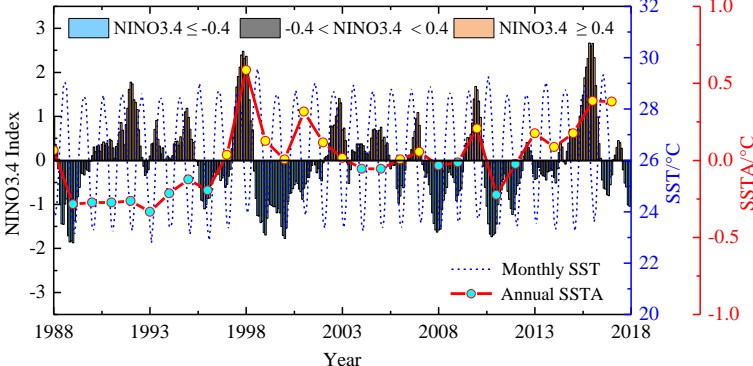


**Figure 5.** The Nino 3.4 index and SST/SSTA during 1988 to 2017. (El Niño in pink and La
Niña in blue.).
It can be seen from Fig.5 that the SSTA minimum value point occurs in 1989 to 1996; the maximum
value point occurs in 1998 and 2016, and the maximum year coincides with the El Niño year. It is shown
that the anomalous changes of the SST in the Northwest Pacific are closely related to the occurrence year
of ENSO. The changes of the SST in the Northwest Pacific are obviously affected by the anomalous
changes of SST in the Equatorial Pacific. The average SSTA was basically negative before 1996, and the
basic value after it was positive. That is, the average SSTA was generally lower than the average of 1988-
2017 before 1996, and the average SSTA after 1996 was basically higher than the average of 1988-2017,
which is also reflected in Fig. 4(c).
In the low-latitude region, SST is more evenly distributed along the latitudes in January to April and
November to December, and are higher in the south and lower in the north. From May to October, the
distribution of SST along the latitude is tilted, showing the distribution characteristics of higher in the
southwest and lower in the northeast, which is affected by the ocean circulation. In addition, as can also
be seen in Fig. 6, in the low-latitude region, the SST range of change in different months is relatively small,
between 27 °C to 33 °C, the change range of 5 °C to 6 °C. In the high-latitude region, the SST can be less
than 3 °C at the lowest, and greater than 15°C at the highest, with a relatively large variation of more than
12 °C.

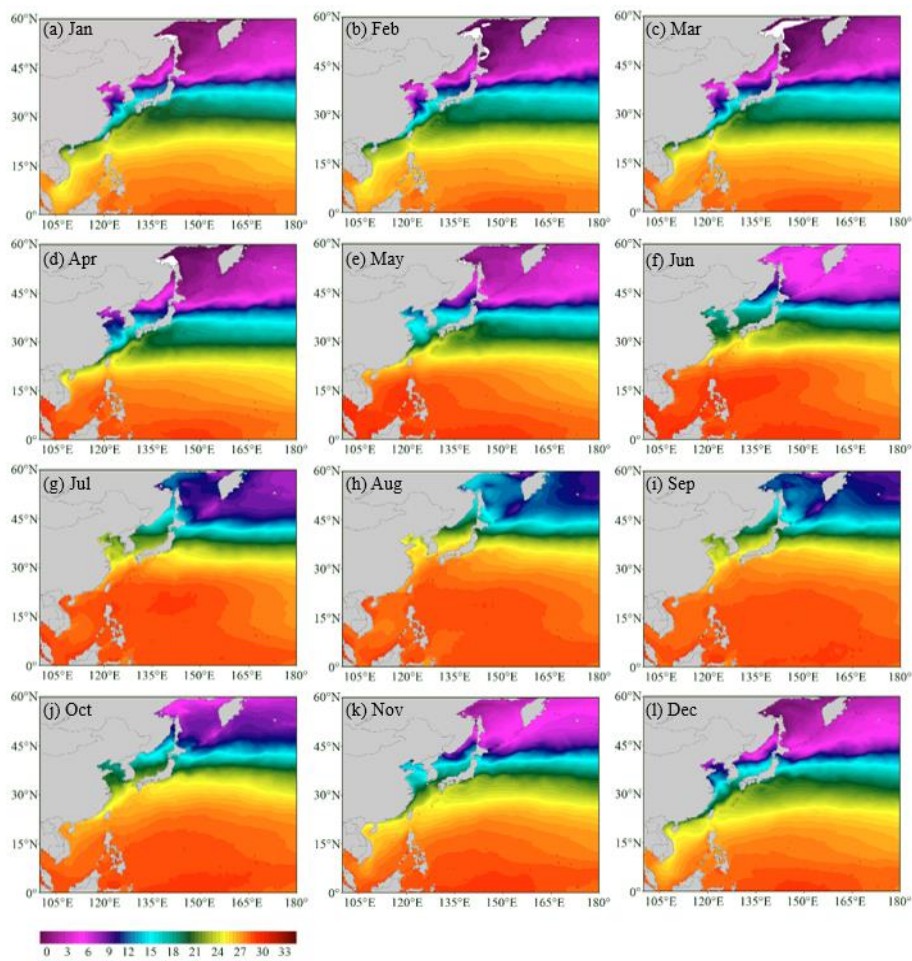

**Figure 6.** Spatial distribution of monthly SST over the 1988-2017 period.

Fig.7 shows the spatial distribution of seasonal and annual mean SST during the 1988-2017 period. As can be seen from the figure, the spatial distribution of average SST in each season and annual is similar, and similar to the monthly results (Fig. 6). In the low-latitude region, the SST is higher, but in the high latitudes. SST is relatively low. Annual mean SST decreases with increasing latitude, with high temperature ranging from 26℃ to 28℃ in the south and low temperature ranging from 3℃ to 6℃ in the north, which is closely related to the solar radiation distribution in the deep-sea region. The isotherm is northeast–southwest oriented and the SST gradient increases as getting closer to the mainland coastal line. It is obvious that the landmass effect in the winter time has contributed to the tilting of the isotherms, which was pointed out by Bao et al (2014).





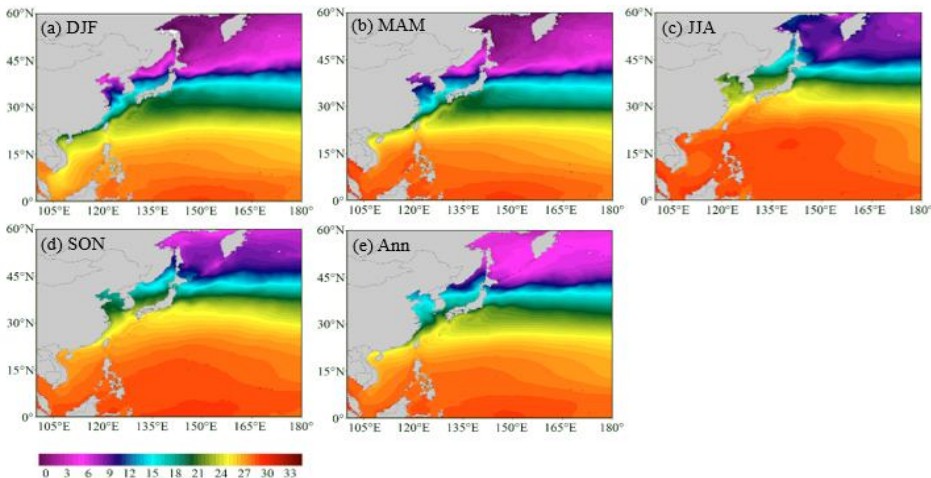


**Figure 7.** Spatial distribution of seasonal/annual SST over the 1988-2017 period (a) Winter:
DJF; (b) Spring: MAM; (c) Summer: JJA; (d) Autumn: SON (e) Annual.
Fig. 8 shows the results of SST anomaly in three characteristic stages. Fig. 8(a) shows the SST
anomaly for the annual 1998 minus 1988-2017, Fig.8 (b) is the annual SST difference between the 10
years after 1998 (1998-2007) and the previous 10 years (1988-1997) and Fig.8 (c) is the SST anomaly for
the last 10 years (2008-2017) and the past 30 years (1988-2017).
It can be seen that there was a significant positive anomaly across the past 30-year average in 1998
from Fig. 8(a). The positive anomalies around 1.0°C are shown in a large area in the Near China Sea,
indicating that the SST is significantly warmer. In the southeast and northeast of the Northwest Pacific,
negative anomalies have occurred in this region, and the lowest is close to -0.6°C, indicating that the SST
has cooled in this region. The SSTA in the Northwest Pacific showed a trend of high in the west and low
in the east. From the previous analysis, we found that this mutation is highly coincident with El Niño (Fig.
5). Therefore, it is likely that this phenomenon has been caused by the temperature difference and time
difference caused by the transfer of high-temperature water in the Northeast Pacific to the Northwest
Pacific.

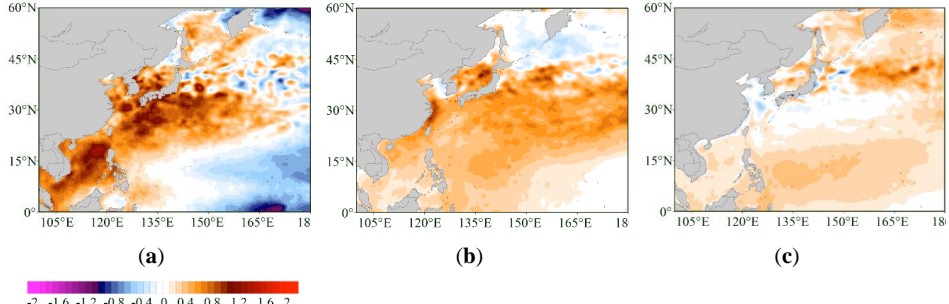


**Figure 8.** (a) Ann 1998 minus 1988-2017; (b) Ann 1998-2007 minus 1988-1997; (c) Ann 2008-
2017 minus 1988-2017.



It can be seen from Fig. 8(b) that the SST during the 10 years from 1998 to 2007 has significantly
increased compared with the previous 10 years from 1988 to 1997. The positive anomaly occurs to be
0.4°C to 0.8°C in the south region of 40° N. In the 10 years since 1998, the SST in the region has increased
by 0.4°C to 0.8°C over the previous 10 years. In the region between 45° N and 60° N, the effect is small
and is maintained between -0.2°C and 0°C, indicating that the SST in this region has not changed
substantially or slightly.
Fig. 8(c) shows the anomalous results of SST over the last 10 years (2008-2017) and relatively nearly
30 years (1988-2017). As can be seen from the figure, in addition to the Bohai Sea, the Yellow Sea, and
the southern region of Japan, there is a wide range of positive anomaly in other regions, and the past 10
years have increased on average in the past 30 years. From Fig. 4(a) and (b), we have known that the
increasing trend of SST over the past 30 years is around three to four times that of the rising trend of SST
over the past 164 years. Therefore, the increasing trend of SST in the past 10 years is more significant,
which is consistent with the results in Fig. 4(c) and Table 1.
*3.3. Correlation between the SST and the atmospheric parameters*
Based on monthly data from ERA-Interim, there is some correlation between SST and atmospheric
parameters have been shown in Fig.9, all marked patterns are at the level of significance equal to 0.05. It
can be seen from Fig. 9(a) that there is a non-significant correlation between SST and North Atlantic
Oscillation (NAO), but in the South China Sea and around the region. It shows a weak negative correlation
between South China Sea SST and NAO. The Pacific Decadal Oscillation (PDO) is an important factor of
climate change of the Northwest Pacific., and it has a strong correlation with ENSO. The PDO has a great
influence on the Asian monsoon and climate change in the Northwest Pacific and is closely related to
ENSO. There is a significant negative correlation between SST and PDO can be seen from Fig. 9(b). The
Niño-3.4 index is usually used to indicate the intensity of the El Niño/La Niña event. So there is a
significant negative correlation between SST and the atmospheric parameters Nino 3.4 in Fig. 9(d).
There is a significant positive correlation between SST and the Southern Oscillation Index (SOI) in
Fig. 9(c), which is a standardized index based on the observed sea level pressure differences between
Tahiti and Darwin, Australia. The monthly correlation between SST and T2 is high throughout the study
region, most markedly (R>0.95) over all Northwest Pacific. The effect of T2 on SST is significant over
98% of the study region in all seasons. This is in good agreement with the previous studies (Skliris et al,
2012; Shaltout and Omstedt, 2014). Similarly, based on monthly data, there is a significant positive
correlation between SST and Total Column Water (TCW), precipitation (PRCP).
The maximum negative correlation between the effect of Wind Speed 10m (WS10) on SST occurs
southeast Northwest Pacific, and significant in an only small region. However, the direct correlation
between V10 and SST is significant and positive over more of the Northwest Pacific.

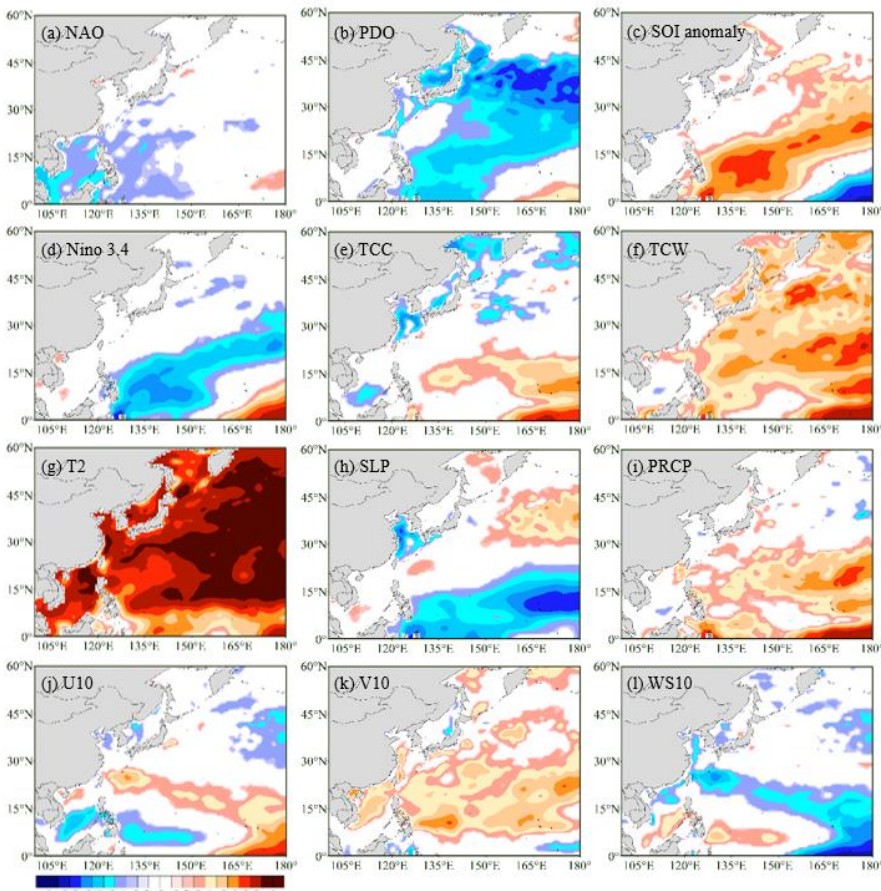


**Figure 9.** The correlation coefficient between SST and the atmospheric components. (level of
significance equal to 0.05).

*3.4. The Near China Sea SST characteristics*

The Near China Sea is defined as the four sea areas of the Bohai Sea, Yellow Sea, East China Sea,
and South China Sea, and include the Kuroshio Extension, the part of Northwest Pacific and the sea
surrounding Japan in this study, which defined as the offshore region of 5°N-41°N and 105°E-130°E. The
changes in the average SST in the Yellow Sea and the Bohai Sea are very similar, so we analyze the two
sea areas together. Therefore, the region is further divided into three sub-regions: Bohai Sea and Yellow
Sea (BYS, 35°N-41°N and 117°E-127°E), East China Sea (ECS, 22°N-35°N and 120°E-130°E) and South
China Sea (SCS, 5°N-22°N and 105°E-120°E) [25].

Fig.11 shows the spatial distribution of seasonal and annual mean SST in the Near China Sea during
the 1988-2017 period. Annual mean SST decreases with increasing latitude, with high temperature ranging
from 26°C to 28°C in the south and low temperature ranging from 14°C to 16°C in the north, which is




closely related to the solar radiation distribution in the offshore region. The isotherm is northeast–
southwest oriented and the SST gradient increases as getting closer to the mainland coastal line. It is
obvious that the landmass effect in the winter has contributed to the tilting of the isotherms, which was
pointed out by Bao et al. [25]. The ECS exhibits the largest temperature gradient, and the SCS in the tropical
zone the lowest temperature gradient.

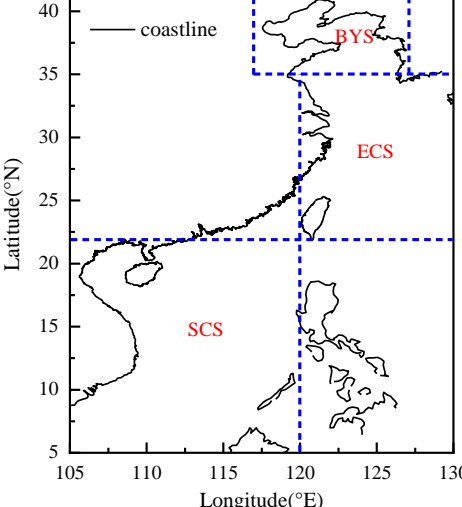


**Figure 10.** Study regions defined in this paper. BYS: the Bohai Sea and the Yellow Sea; ECS:
the East China Sea; SCS: the South China Sea.

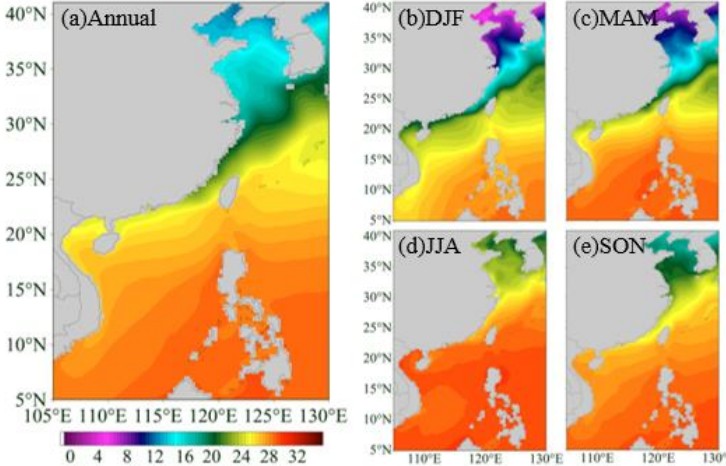


**Figure 11.** Annual (left) and seasonal (right) mean SST distribution during 1988-2017 in the
China Sea. (a) Annual; (b) Winter: DJF; (c) Spring: MAM; (d) Summer: JJA; (e) Autumn: SON.





The monthly mean surface temperature changes over the past 10 years in the three regions (BYS,
ECS and SCS) and the whole sea area (China Sea) are shown in Fig. 12. Fig. 12(a) shows the year-by-year
variation of SST in different regions in the last 10 years, and Fig.12(b) shows the monthly SST variations
in different regions in the past 10 years. The change variability of SST in different regions are basically
synchronized. The minimum temperature basically occurs in February and the warmest occurs in August.
The fluctuation range of SST in BYS is the largest, basically between 5 °C to 22 °C, from 18 °C to 27 °C
in the East China Sea, and the smallest fluctuations is in the South China Sea, maintained at a range of
26 °C to 29 °C. There are large differences between the mean and standard deviation in different regions.

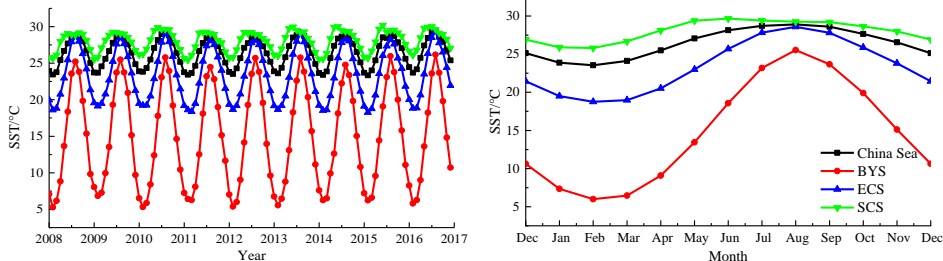


**Figure 12.** Long term monthly mean SST of the marginal seas of China during 2008-2017 (a)
Yearly; (b) Monthly. Black line: China Sea; red line: Bohai Sea and Yellow Sea (BYS); blue
line: East China Sea (ECS); green line: South China Sea (SCS).
Table 2 shows the annual and seasonal SST characteristics of the study area Near China Sea based
on monthly data from 1988 to 2017. It can be found that in addition to the winter and spring in the BYS,
the SST in each season of other regions shows an increasing trend from the table. Average increasing
trends of SST during 1988 to 2017 in BYS is 0.015 °C/ 10yr, 0.14 °C/ 10yr for the ECS, 0.12 °C/ 10yr for
the SCS and 0.12 °C/ 10yr for whole Near China Sea respectively, and all the trends are significant at the
99% confidence level. From the point of average annual SST, the SST in the South China Sea is the highest,
reaching 28.01°C, followed by the East China Sea with 23.4°C, the lowest in the Bohai Sea and the Yellow
Sea is 14.98°C, and the SST in the whole Near China Sea is 26.4°C. Table 3 shows the peak value and
time of the annual and seasonal SST of the study area Near China Sea based on monthly data from 1988
to 2017. In the past 30 years, colder SST occurs in 1989, 1990, 1992, 1993, 2003, 2008, 2010, 2011.
Warmer SST occurs in 1997, 1998, 1999, 2001, 2015, 2016.





Table 2. Annual and seasonal SST characteristics of the study area Near China Sea based on monthly data from 1988 to 2017.

| | Average trend (°C/10yr) | | | | | Average (°C) ± standard deviation | | | | |
|---|---|---|---|---|---|---|---|---|---|---|
| | Winter | Spring | Summer | Autumn | Annual | Winter | Spring | Summer | Autumn | Annual |
| BYS | -0.027 | -0.097 | 0.084 | 0.13 | **0.015** | 8.08 ± 0.52 | 9.84 ± 0.49 | 22.44 ± 0.54 | 19.56 ± 0.44 | **14.98 ± 0.34** |
| ECS | 0.11 | 0.04 | 0.15 | 0.23 | **0.14** | 19.81 ± 0.33 | 20.87 ± 0.35 | 27.24 ± 0.31 | 25.66 ± 0.34 | **23.40 ± 0.26** |
| SCS | 0.13 | 0.10 | 0.11 | 0.14 | **0.12** | 26.09 ± 0.33 | 28.02 ± 0.27 | 29.38 ± 0.28 | 28.54 ± 0.27 | **28.01 ± 0.23** |
| Whole | 0.13 | 0.08 | 0.11 | 0.16 | **0.12** | 24.07 ± 0.27 | 25.53 ± 0.25 | 28.50 ± 0.24 | 27.50 ± 0.26 | **26.40 ± 0.21** |

Table 3. Peak value and time of the annual and seasonal SST of the study area Near China Sea based on monthly data from 1988 to 2017.

| | Minimum (°C) and time (yr) | | | | | Maximum (°C) and time (yr) | | | | |
|---|---|---|---|---|---|---|---|---|---|---|
| | Winter | Spring | Summer | Autumn | Annual | Winter | Spring | Summer | Autumn | Annual |
| BYS | 7.13 (2003) | 8.88 (2010) | 21.13 (1993) | 18.69 (1992) | 14.45 (2010) | 9.17 (2001) | 11.02 (1998) | 23.99 (1997) | 20.70 (1998) | 15.85 (1998) |
| ECS | 19.30 (1989) | 20.04 (2011) | 26.76 (1993) | 25.01 (1992) | 22.97 (1993) | 20.54 (1999) | 21.84 (1998) | 28.06 (2016) | 26.43 (1998) | 24.14 (1998) |
| SCS | 25.53 (1993) | 27.50 (2011) | 28.97 (2008) | 27.98 (1992) | 27.68 (1989) | 26.78 (2016) | 28.53 (2001) | 30.02 (1998) | 29.14 (2015) | 28.58 (1998) |
| Whole | 23.61 (1993) | 24.99 (2011) | 28.18 (1990) | 26.94 (1992) | 26.07 (1993) | 24.63 (1999) | 26.05 (1998) | 29.09 (1998) | 28.18 (1998) | 26.98 (1998) |

## 4. Conclusions

The Northwest Pacific sea surface variability is affected by a combination of oceanic and atmospheric processes and displays significant regional and seasonal behavior. Monthly SST datasets based on ERSST 3b (1854-2017, 164 years) and OISST V2 (1988-2017, 30 years) are used to make some long-term temporal and spatial variability statistics. The following conclusions can be drawn from the analysis.

In the last 164 years, SST in the Northwest has gradually increased, with an increasing trend of 0.033 °C/10 yr. Especially in the past 30 years, the increasing trend of SST reaches to 0.132 °C/10 yr, and the increasing trend of SST reaches to 0.306 °C/10 yr in the last 10 years, which increasing trend is very obviously. The trend of the SST varies seasonally. The increasing trend in winter and autumn are 0.124 °C/10 yr and 0.146 °C/10 yr respectively, which are greater than spring and summer, with 0.075 °C/10 yr and 0.107°C /10 yr respectively. There was an SST mutation point occurred around 1998, the average annual SST for the 10 years after 1998 increased by 0.3°C over the previous 10 years. It has been found that the change of SST/SSTA in the Northwest Pacific is closely related to the ENSO through the statistical analysis of Nino3.4 index and SST/SSTA.


From the perspective of spatial distribution, the annual mean SST decreases with increasing latitude

in conclusion, with high temperature ranging from 27°C to 33°C in the south and low temperature ranging
from 3°C to 15°C in the north. The SST is higher in the low-latitude (near equator) region and lower in
the high-latitude region. In the low-latitude region, SST is more evenly distributed along the latitudes in
November to April, but from May to October, the distribution of SST along the latitude is tilted, showing
the distribution characteristics of higher in the southwest and lower in the northeast, which is affected by
the ocean circulation.

There are many correlations between the SST and some climate indexes and atmospheric parameters,

such as Pacific Decadal Oscillation (PDO), Southern Oscillation Index (SOI), Nino 3.4, total water vapor
column (TWC), temperature at 2 meters (T2), sea level pressure (SLP), precipitation (PRCP) and wind
speed at 10 meters (U10, V10 and WS10). A very significant positive correlation between SST and T2,
TCW was been found, of which the correlation coefficient between SST and T2 exceeded 98%. PDO,
Nino 3.4 is negatively correlated with SST, and the correlation between other indexes and parameters and
SST is weak.

The whole Near China Sea was divided into three sections to analysis its spatial variability in a

different region, which is the Bohai Sea and Yellow Sea (BYS), East China Sea (ECS) and South China
Sea (SCS). The SST in the BYS is coolest with a range from 5 °C to 22 °C, and the warmest in the SCS
with a range from 26 °C to 29 °C. It can be seen from the statistical data that in addition to the winter and
spring in the BYS, SST in other regions and time had shown a warming trend. In the past 30 years, the
trend of SST increase of BYS was 0.015 °C/10 yr, while that of ECS and SCS was 0.14 °C/10 yr and
0.12 °C/10 yr, respectively.
**Competing interests:** The authors declare that they have no conflict of interest.
**Financial support:** The study was supported by the National Natural Science Foundation of China
(Grant Nos. 51809023, 51839002 and 51879015).

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
