# Peer review of "The long-term spatio-temporal variability of sea surface 2 temperature in the Northwest Pacific and the Near China Sea"

_Ocean Science, 2019_

## Editor Comment (EC1) · Neil Wells (Editor) · 18 Sep 2019

OS-2019 -69

The long term spatio-temporal variability of sea surface temperature in the NW Pacific and near China Sea

Zhiyuan Wu et. al.

General Comments

This paper describes the analysis of trends in a long SST time series in the NW Pacific and relates this sub-regions near the Chinese mainland and other sub –regions in NW Pacific. Furthermore it relates the SST to some climate indices. This should have potential interest among many people in the climate community. However in its present form it will need a substantial revision before it is accepted for publication.   I have detailed below my comments on the paper. In your reply please give specific answers to each major comment.

Major comments

Line 134-136 I am not convinced this statement is correct as it stands. HADISST is a long term data set 1850-present.  Need to say more about your reasons for using the data set you used.

Line 162 The ECMWF produces 10 day global forecasts and it certainly doesn't focus on mesoscale weather forecasting ( very high resolution regional forecasts).

Line 229 -230 This sentence is not clear. What does the curve trend is very gentle mean. ? What does oscillated gradually mean ?  Also the SST is the valley of nearly 164 years should be expressed perhaps as the SST is at a minimum over the 164 years.

Line 243 – 244 You do not explain why ±0.4 °C is used for discriminating anomalies. Is it 1 standard deviation of the time series or is it the tercile value?  Your statistics could be biased if you did not use the correct boundary.

Line 253-258 You use a term " mutation" which is not used in European oceanography or meteorology because it is widely term used in biological sciences. You need to replace it with a more appropriate word or words throughout your paper.

Line 281-288 A correlation coefficient (with significance level) with ENSO index should be given here.

A figure reference should also be added in this paragraph.

Line 321-323You need to explain how high temperature water can be transferred from the NE Pacific to NW Pacific. It may not necessarily be transferred by the ocean circulation. The atmosphere circulation does play a role by ocean-air transfer from the ENSO region.

Line 333-339 A linear regression has been used throughout the paper. But clearly the time series is non-linear in the later part of the data set. This would suggest either non-linear regression or a low order polynomial may be  more suitable to describe the series. ?

Line p341-360 The correlation maps shown in Figure 9 are very interesting but the discussion of these maps needs to improved. For example there is a brief mention of significance when discussing SST and T2 but not in any other of the  correlations shown in figure 9. In particular the SST and ENSO doesn't give a significance level for the correlation map.

A further point about this discussion is the mention that PDO and ENSO are significantly correlated but this map is not shown in figure 9. If it is well known they are correlated then a reference is needed.

This section of the paper needs to be revised carefully to discuss each correlation map in Figure 9 with levels of significance given.

Line 362 Figure 9 The abbreviations such as TCC, TCW and PRCP have not been defined in the methods section on p4 and p5. They should **all** be defined e.g. precipitation (PRCP) in the methods section.

Line 426-428 Not convinced this has been demonstrated in Section 3.3 ( p341-360).

Line 429-435 The description of seasonal temperature distribution ( May to October) refers to ocean circulation being the cause of the tilted distribution but again no evidence is supplied or a reference given. It could be result of upwelling at the coastal boundary.

Minor Comments

Line 19-20 The sentence should be made clearer. A slow decreasing trend period does not make any sense to me. Also a trough in the time series is not appropriate scientific language in this context. You should state " 1910-1930 was the lowest  minimum in the 164 year record. "

Line 24 Should be "The change in trend"

Line 43 Should this be "Ocean heat content" …and dynamic processes.

Line 59 add a comma after " droughts" and remove " and"

Line 92  Replace "space " by "research"

Line 116 Replace " are " by "is"

Line 153 Replace "in the north" by " to the north"

Line 189 Replace "Perform a significance test"  by "A significance test is performed…"

Line 210 Figure 3 (top graph) I was surprised that the domain covers 0-60N with temperatures ranging from 3-6C in the north to 26 to 28 C but the mean is about 26 C ? Need to check this is correct.

Line 213 Legend " All the trends are significant" not " is  siginificant.

Line 218 Should be North Western Pacific"

Line 225 Should be " 95% significance test "

Line 238 Should be " red lines are their trends"

Line 239 -240 I suggest removing "The same as the annual pattern, seasonal pattern" replacing by " The seasonal pattern for the latest 30 years shows a more significant warming trend than that over the 164 year period."

Line 243 Insert " is" after anomaly.

Line 251 Delete " curve"

Line 252 Replace "was" by " is"

Line 266 Figure 4 Insert (c) for lower part of figure

Line 285 Remove " basically"

Line 287 Remove " basically"

Line 289-296 Need a figure reference in this section ( ie Figure 6 and/or 7)

Line 305 I suggest replacing "getting closer to the main land coast line" by " it approaches the coast"

Line 333 Remove "relatively nearly" as it confuses the meaning of the sentence.

Line 335 Replace " anomaly" by " anomalies"

Line 338 You need show the significance level for the 10 year period. This is important because there is variability in the climate record at the decadal time scale.

Line 366-368 Sentence is confusing. Suggest  adding a full stop after South China Sea. And Starting a new sentence "The Kuroshio Extension …, is defined as the offshore region…"

Line 377 Replace "SST gradient increases as getting closer to the mainland coast line" by "SST gradient increases as it approaches the mainland coast"

Line 379 Year needed for Bao et al reference

Line 388 Should be " Near China Sea" in brackets to be consistent.

Line 440 –line 441 Should be " PDO and Nino 3.4"

---

## Referee Comment (RC1) · Anonymous Referee #1 · 20 Sep 2019

The manuscript "The long-term spatio-temporal variability of sea surface temperature in the Northwest Pacific and the Near China Sea" by Zhiyuan Wu et al., Presents the variability of the sea surface temperature (SST) in Northwest Pacific the last 164 years, on seasonal, annual and interannual scales based on monthly data sets. The analysis is well presented, and the results are interesting in terms of global warming. The correlations found are mostly expected, especially between the SST and the T2, since the temperature at 2 m and SST are strongly linked. The changes in the SST and the SSTA are closely related to El Niño 3.4. The important part of this study is the increasing SST linear trend of 0.033 °C/10 yr and especially of 0.306 °C/10 yr in the last ten years, which shows an "acceleration" in the temperature increase in the

[Figure]

Northwest Pacific.

On the other hand, it is interesting the change that the authors find in the SST around 1998. Although they do not propose an explanation for this change, it would be excellent if they tried to give some comment or proposed a hypothesis. A minor issue is in Figure 12, placing a, and b on the figures to be consistent with the figure caption. The manuscript deserves to be accepted.

––––––––––––––––––––––––––––

---

## Author Comment (AC1) · 22 Nov 2019

Response to comments of Reviewers
The authors are grateful to this reviewer for pin-point and pertinent comments and checking the paper. All comments are addressed point by point, each starting with an original comment and followed by a response in italic, as follows.

[Figure]

The manuscript "The long-term spatio-temporal variability of sea surface temperature in the Northwest Pacific and the Near China Sea" by Zhiyuan Wu et al., Presents the variability of the sea surface temperature (SST) in Northwest Pacific the last 164 years, on seasonal, annual and interannual scales based on monthly data sets. The analysis is well presented, and the results are interesting in terms of global warming. Response: Thank you for these comments. The positive comments in our solid professional skills are good encouragement to us.

The correlations found are mostly expected, especially between the SST and the T2, since the temperature at 2 m and SST are strongly linked. The changes in the SST and the SSTA are closely related to El Niño 3.4. The important part of this study is the increasing SST linear trend of 0.033 âŮęC/10 yr and especially of 0.306 âŮęC/10 yr in the last ten years, which shows an "acceleration" in the temperature increase in the Northwest Pacific. Response: We are grateful to these positive comments.

On the other hand, it is interesting the change that the authors find in the SST around 1998. Although they do not propose an explanation for this change, it would be excellent if they tried to give some comment or proposed a hypothesis. Response: Thank you for your comment. As the reviewer said, we found a very interesting phenomenon about the changes in the SST around 1998. We believe that this phenomenon has an important relationship with the El Niño, which can be confirmed in Figure 5. And the same phenomenon reappeared around 2016, we can understand this phenomenon from the NINO3.4 index. These discussions are in Section 3.1, lines 269 to 288 of the manuscript.

A minor issue is in Figure 12, placing a, and b on the figures to be consistent with the figure caption. Response: Thanks for your careful checking. We revised the Figure 12.

The manuscript deserves to be accepted. Response: We are grateful to the positive comment and encouragement.

[Figure]

Please also note the supplement to this comment:
https://www.ocean-sci-discuss.net/os-2019-69/os-2019-69-AC1-supplement.pdf

**Supplement:**

*The authors are grateful to this reviewer for pin-point and pertinent comments and checking the paper. All comments are addressed point by point, each starting with an original comment and followed by a response in italic, as follows.*

The manuscript "The long-term spatio-temporal variability of sea surface temperature in the Northwest Pacific and the Near China Sea" by Zhiyuan Wu et al., Presents the variability of the sea surface temperature (SST) in Northwest Pacific the last 164 years, on seasonal, annual and interannual scales based on monthly data sets. The analysis is well presented, and the results are interesting in terms of global warming.

*Response: Thank you for these comments. The positive comments in our solid professional skills are good encouragement to us.*

The correlations found are mostly expected, especially between the SST and the T2, since the temperature at 2 m and SST are strongly linked. The changes in the SST and the SSTA are closely related to El Niño 3.4. The important part of this study is the increasing SST linear trend of 0.033 $\circ$C/10 yr and especially of 0.306 $\circ$C/10 yr in the last ten years, which shows an "acceleration" in the temperature increase in the Northwest Pacific.

*Response: We are grateful to these positive comments.*

On the other hand, it is interesting the change that the authors find in the SST around 1998. Although they do not propose an explanation for this change, it would be excellent if they tried to give some comment or proposed a hypothesis.

*Response: Thank you for your comment. As the reviewer said, we found a very interesting phenomenon about the changes in the SST around 1998. We believe that this phenomenon has an important relationship with the El Niño, which can be confirmed in Figure 5. And the same*

*phenomenon reappeared around 2016, we can understand this phenomenon from the NINO3.4*

*index. These discussions are in Section 3.1, lines 269 to 288 of the manuscript.*

A minor issue is in Figure 12, placing a, and b on the figures to be consistent with the figure caption.

*Response: Thanks for your careful checking. We revised the Figure 12.*

The manuscript deserves to be accepted.

*Response: We are grateful to the positive comment and encouragement.*

[revised manuscript text omitted]

---

## Author Comment (AC2) · 22 Nov 2019

**Please check the attachment for a better reading experience.

Response to comments from Editor
Dr. Neil Wells knows the topic very well and his careful checking and constructive comments are indeed helpful in improving the quality of our manuscript. We are grateful

to Dr. Wells for his patience. All comments are addressed point by point, each starting with an original comment and followed by a response in italic, as follows.

General Comments This paper describes the analysis of trends in a long SST time series in the NW Pacific and relates this sub-regions near the Chinese mainland and other sub –regions in NW Pacific. Furthermore it relates the SST to some climate indices. This should have potential interest among many people in the climate community. However in its present form it will need a substantial revision before it is accepted for publication. I have detailed below my comments on the paper. In your reply please give specific answers to each major comment. Response: We are grateful to these positive comments and encouragement, and we are also grateful to the pin-point and pertinent comments and checking the paper.

Major comments Line 134-136 I am not convinced this statement is correct as it stands. HADISST is a long term data set 1850-present. Need to say more about your reasons for using the data set you used. Response: Thanks for your comment. This opinion is recognized by some scholars, such as Kim et al (2018). But after you disagree, I found this statement to be inaccurate. The HadISST1 data set replaces the GISST data sets, and is a unique combination of monthly globally-complete fields of SST and sea ice Concentration on a 1 degree latitude-longitude grid from 1870 to date. Fields for the month-before-last are added to the data set on the 2nd of every new month. But, from May 2007 the data set of in situ measurements used in HadISST has changed. The MOHSST data set, which was previously used has been discontinued, and HadSST2 is now being used in its place. We added this reasons in the revised manuscript.

Kim Y S, Jang C J, Yeh S W. Recent surface cooling in the Yellow and East China Seas and the associated North Pacific climate regime shift[J]. Continental Shelf Research, 2018, 156: 43-54.

Line 162 The ECMWF produces 10 day global forecasts and it certainly doesn't focus on mesoscale weather forecasting (very high resolution regional forecasts). Response:

Thanks for your comment. This is indeed an important conceptual error. The goal of the center is to release a customized mid-term weather forecasting (temporal) not mesoscale weather forecasting (spatial). We corrected it in the revised manuscript.

Line 229 -230 This sentence is not clear. What does the curve trend is very gentle mean. ? What does oscillated gradually mean? Also the SST is the valley of nearly 164 years should be expressed perhaps as the SST is at a minimum over the 164 years. Response: Thanks for your comment and suggestion. These sentences were rewritten as following and we hope it is more readable. It can be seen from Fig. 3 that during the period of 1870-1910, the SST slowly decreased, staying in the range between 25.2 °C to 26.0 °C; during the period of 1910-1930, the SST as whole maintained a low value, and the change range was small, which is at the minimum over the 164 years; since 1930, the SST has started to rise with oscillation and the trend has continued to this day.

Line 243 - 244 You do not explain why $\pm 0.4$ °C is used for discriminating anomalies. Is it 1 standard deviation of the time series or is it the tercile value? Your statistics could be biased if you did not use the correct boundary. Response: Thanks for your comment and suggestion. An El Niño or La Niña event is identified if the 5-month running-average of the NINO3.4 index exceeds +0.4°C for El Niño or -0.4°C for La Niña for at least 6 consecutive months, so $\pm 0.4$ °C is used for discriminating anomalies in this study. We added this explanation in the revised manuscript.

Line 253-258 You use a term "mutation" which is not used in European oceanography or meteorology because it is widely term used in biological sciences. You need to replace it with a more appropriate word or words throughout your paper. Response: Thanks for your professional comment. We used the term "extremum" (or "extreme") instead of "mutation" in the revised manuscript.

Line 281-288 A correlation coefficient (with significance level) with ENSO index should be given here. A figure reference should also be added in this paragraph. Response:

Thank you for your suggestion. Since it is not clear whether SSTA is related to ENSO index, the correlation coefficient SSTA with ENSO index had not be given here. What is emphasized here is that El Nino phenomenon will lead to obvious changes in SSTA, which can be shown in Figure 5.

Line 321-323 You need to explain how high temperature water can be transferred from the NE Pacific to NW Pacific. It may not necessarily be transferred by the ocean circulation. The atmosphere circulation does play a role by ocean-air transfer from the ENSO region. Response: Thank you for your comment and suggestion. The heat transfer here is not only the result of the ocean circulation, but also the result of the interaction between the ocean and the atmosphere, including the relationship between the Walker Circulation and El Niño, and the combination of atmospheric circulation and ocean circulation. We corrected it in the revised manuscript.

Line 333-339 A linear regression has been used throughout the paper. But clearly the time series is non-linear in the later part of the data set. This would suggest either non-linear regression or a low order polynomial may be more suitable to describe the series. ? Response: Thanks for your professional comment. From the perspective of similarity fitting or mathematics, as you said, the accuracy may be higher with non-linear regression or a low order polynomial. However, from the perspective of trend comparison, the linear fitting method can reflect the results more intuitively.

Line p341-360 The correlation maps shown in Figure 9 are very interesting but the discussion of these maps needs to improved. For example there is a brief mention of significance when discussing SST and T2 but not in any other of the correlations shown in figure 9. In particular the SST and ENSO doesn't give a significance level for the correlation map. A further point about this discussion is the mention that PDO and ENSO are significantly correlated but this map is not shown in figure 9. If it is well known they are correlated then a reference is needed. Response: Thanks for your comment. Some correlation between SST and atmospheric parameters at the level of significance equal to 0.05 have been shown in Figure 9. All the discussion, such as
the correlations SST and T2, SST and ENSO, PDO and ENSO, is based on the level of significance equal to 0.05. The discussion about the PDO and ENSO is based on the Figure 9 (b), (c) and (d).

Line 362 Figure 9 The abbreviations such as TCC, TCW and PRCP have not been defined in the methods section on p4 and p5. They should all be defined e.g. precipitation (PRCP) in the methods section. Response: Thanks for your suggestion. The abbreviations such as TCC, TCW and PRCP indeed have not been defined in the previous section. However, we used the full name when first used the abbreviation in the text, such as "Total Column Water (TCW), precipitation (PRCP)". So it should not affect the understanding of the discussion.

Line 426-428 Not convinced this has been demonstrated in Section 3.3 ( p341-360). Response: Thanks for your comment. The conclusion that the change of SST/SSTA in the Northwest Pacific is closely related to the ENSO through the statistical analysis of Nino3.4 index and SST/SSTA is based not only on Section 3.3 (Line 341-360), but also on the conclusions of the discussion in Sections 3.1 and 3.3.

Line 429-435 The description of seasonal temperature distribution (May to October) refers to ocean circulation being the cause of the tilted distribution but again no evidence is supplied or a reference given. It could be result of upwelling at the coastal boundary. Response: Thanks for your professional comment. We added the relevant explanation before Figure 6.

Minor Comments Line 19-20 The sentence should be made clearer. A slow decreasing trend period does not make any sense to me. Also a trough in the time series is not appropriate scientific language in this context. You should state "1910-1930 was the lowest minimum in the 164 year record." Response: Thanks for your suggestion. We revised it.

Line 24 Should be "The change in trend" Response: Thank you for the suggestion. We corrected it.

Line 43 Should this be "Ocean heat content" . . .and dynamic processes. Response: Thank you for the suggestion. We corrected it.

Line 59 add a comma after " droughts" and remove " and" Response: Thank you for the suggestion. We corrected it.

Line 92 Replace "space " by "research" Response: Thank you for the suggestion. We corrected it.

Line 116 Replace " are " by "is" Response: Thank you for the suggestion. We corrected it.

Line 153 Replace "in the north" by " to the north" Response: Thank you for the suggestion. We corrected it.

Line 189 Replace "Perform a significance test" by "A significance test is performed. . ." Response: Thank you for the suggestion. We corrected it.

Line 210 Figure 3 (top graph) I was surprised that the domain covers 0-60N with temperatures ranging from 3-6C in the north to 26 to 28 C but the mean is about 26 C ? Need to check this is correct. Response: Thank you for the suggestion. We corrected it.

Line 213 Legend " All the trends are significant" not " is siginificant. Response: Thank you for the suggestion. We corrected it.

Line 218 Should be North Western Pacific" Response: Thank you for the suggestion. We corrected it.

Line 225 Should be " 95% significance test " Response: Thank you for the suggestion. We corrected it.

Line 238 Should be " red lines are their trends" Response: Thank you for the suggestion. We corrected it.

Line 239 -240 I suggest removing "The same as the annual pattern, seasonal pattern" replacing by " The seasonal pattern for the latest 30 years shows a more significant warming trend than that over the 164 year period." Response: Thank you for the suggestion. We corrected it.

Line 243 Insert " is" after anomaly. Response: Thank you for the suggestion. We corrected it.

Line 251 Delete " curve" Response: Thank you for the suggestion. We corrected it.

Please also note the supplement to this comment:
https://www.ocean-sci-discuss.net/os-2019-69/os-2019-69-AC2-supplement.pdf

———————————————

[Figure]

**Supplement:**

Editor comment

*Dr. Neil Wells knows the topic very well and his careful checking and constructive comments are indeed helpful in improving the quality of our manuscript. We are grateful to Dr. Wells for his patience. All comments are addressed point by point, each starting with an original comment and followed by a response in italic, as follows.*

**General Comments**

This paper describes the analysis of trends in a long SST time series in the NW Pacific and relates this sub-regions near the Chinese mainland and other sub –regions in NW Pacific. Furthermore it relates the SST to some climate indices. This should have potential interest among many people in the climate community. However in its present form it will need a substantial revision before it is accepted for publication. I have detailed below my comments on the paper. In your reply please give specific answers to each major comment.

*Response: We are grateful to these positive comments and encouragement, and we are also grateful to the pin-point and pertinent comments and checking the paper.*

**Major comments**

Line 134-136 I am not convinced this statement is correct as it stands. HADISST is a long term data set 1850-present. Need to say more about your reasons for using the data set you used.

*Response: Thanks for your comment. This opinion is recognized by some scholars, such as Kim et al (2018). But after you disagree, I found this statement to be inaccurate. The HadISST1 data set replaces the GISST data sets, and is a unique combination of monthly globally-complete fields of SST and sea ice Concentration on a 1 degree latitude-longitude grid* ***from 1870 to date****. Fields for the month-before-last are added to the data set on the 2nd of every new month.* ***But, from May 2007 the data set of in situ measurements used in HadISST***

***has changed.*** *The MOHSST data set, which was previously used has been discontinued, and HadSST2 is now being used in its place. We added this reasons in the revised manuscript.*

Y.S. Kim et al.

data/gridded/data.noaa.oisst.v2.html). A monthly mean SST dataset for the period 1982–2014 (i.e., 33 years) was used in this study. Considering that the study area of the YECS covers only 12.5° longitudes by 12° latitudes (i.e., 117–129.5°E, 29–41°N), we believe that the OISSTv2 is the most suitable SST dataset for this study due to its fine spatial resolution (i.e., 1/4° × 1/4°) without degrading or a systematic bias for more than three decades (e.g., Reynolds and Chelton, 2010). The advantage of this dataset is apparent when compared with other gridded datasets such as the Hadley Center Ice and Sea Surface Temperature (HadISST; 1° horizontal resolution), the Extended Reconstructed Sea Surface Temperature version 4 (ERSSTv4; 2° resolution), and the Operational Sea Surface Temperature and Sea Ice Analysis (OSTIA, 1/20° resolution), which spans only the period since 2007.

*Kim Y S, Jang C J, Yeh S W. Recent surface cooling in the Yellow and East China Seas and the associated North Pacific climate regime shift[J]. Continental Shelf Research, 2018, 156: 43-54.*

Line 162 The ECMWF produces 10 day global forecasts and it certainly doesn't focus on mesoscale weather forecasting (very high resolution regional forecasts).

    *Response: Thanks for your comment. This is indeed an important conceptual error. The goal of the center is to release a customized mid-term weather forecasting (temporal) not mesoscale weather forecasting (spatial). We corrected it in the revised manuscript.*

Line 229 -230 This sentence is not clear. What does the curve trend is very gentle mean. ? What does oscillated gradually mean? Also the SST is the valley of nearly 164 years should be expressed perhaps as the SST is at a minimum over the 164 years.

    *Response: Thanks for your comment and suggestion. These sentences were rewritten as following and we hope it is more readable. It can be seen from Fig. 3 that during the period of 1870-1910, the SST slowly decreased, staying in the range between 25.2 °C to 26.0 °C; during the period of 1910-1930, the SST as whole maintained a low value, and the change range was small, which is at the minimum over the 164 years; since 1930, the SST has started to rise with oscillation and the trend has continued to this day.*

Line 243 - 244 You do not explain why ±0.4 °C is used for discriminating anomalies. Is it 1 standard deviation of the time series or is it the tercile value? Your statistics could be biased if you did not use the correct boundary.

*Response: Thanks for your comment and suggestion. An El Niño or La Niña event is identified if the 5-month running-average of the NINO3.4 index exceeds +0.4°C for El Niño or -0.4°C for La Niña for at least 6 consecutive months, so ±0.4 °C is used for discriminating anomalies in this study. We added this explanation in the revised manuscript.*

Line 253-258 You use a term "mutation" which is not used in European oceanography or meteorology because it is widely term used in biological sciences. You need to replace it with a more appropriate word or words throughout your paper.

*Response: Thanks for your professional comment. We used the term "extremum" (or "extreme") instead of "mutation" in the revised manuscript.*

Line 281-288 A correlation coefficient (with significance level) with ENSO index should be given here. A figure reference should also be added in this paragraph.

*Response: Thank you for your suggestion. Since it is not clear whether SSTA is related to ENSO index, the correlation coefficient SSTA with ENSO index had not be given here. What is emphasized here is that El Nino phenomenon will lead to obvious changes in SSTA, which can be shown in Figure 5.*

Line 321-323 You need to explain how high temperature water can be transferred from the NE Pacific to NW Pacific. It may not necessarily be transferred by the ocean circulation. The atmosphere circulation does play a role by ocean-air transfer from the ENSO region.

*Response: Thank you for your comment and suggestion. The heat transfer here is not only the result of the ocean circulation, but also the result of the interaction between the ocean and the atmosphere, including the relationship between the Walker Circulation and El Niño, and the combination of atmospheric circulation and ocean circulation. We corrected it in the revised manuscript.*

Line 333-339 A linear regression has been used throughout the paper. But clearly the time series is non-linear in the later part of the data set. This would suggest either non-linear regression or a low order polynomial may be more suitable to describe the series. ?

*Response: Thanks for your professional comment. From the perspective of similarity fitting or mathematics, as you said, the accuracy may be higher with non-linear regression or a low order polynomial. However, from the perspective of trend comparison, the linear fitting method can reflect the results more intuitively.*

Line p341-360 The correlation maps shown in Figure 9 are very interesting but the discussion of these maps needs to improved. For example there is a brief mention of significance when discussing SST and T2 but not in any other of the correlations shown in figure 9. In particular the SST and ENSO doesn't give a significance level for the correlation map.

A further point about this discussion is the mention that PDO and ENSO are significantly correlated but this map is not shown in figure 9. If it is well known they are correlated then a reference is needed.

*Response: Thanks for your comment. Some correlation between SST and atmospheric parameters at the level of significance equal to 0.05 have been shown in Figure 9. All the discussion, such as the correlations SST and T2, SST and ENSO, PDO and ENSO, is based on the level of significance equal to 0.05. The discussion about the PDO and ENSO is based on the Figure 9 (b), (c) and (d).*

Line 362 Figure 9 The abbreviations such as TCC, TCW and PRCP have not been defined in the methods section on p4 and p5. They should all be defined e.g. precipitation (PRCP) in the methods section.

*Response: Thanks for your suggestion. The abbreviations such as TCC, TCW and PRCP indeed have not been defined in the previous section. However, we used the full name when first used the abbreviation in the text, such as "Total Column Water (TCW), precipitation (PRCP)". So it should not affect the understanding of the discussion.*

Line 426-428 Not convinced this has been demonstrated in Section 3.3 ( p341-360).

*Response: Thanks for your comment. The conclusion that the change of SST/SSTA in the Northwest Pacific is closely related to the ENSO through the statistical analysis of Nino3.4 index and SST/SSTA is based not only on Section 3.3 (Line 341-360), but also on the conclusions of the discussion in Sections 3.1 and 3.3.*

Line 429-435 The description of seasonal temperature distribution (May to October) refers to ocean circulation being the cause of the tilted distribution but again no evidence is supplied or a reference given. It could be result of upwelling at the coastal boundary.

*Response: Thanks for your professional comment. We added the relevant explanation before Figure 6.*

**Minor Comments**

Line 19-20 The sentence should be made clearer. A slow decreasing trend period does not make any sense to me. Also a trough in the time series is not appropriate scientific language in this context. You should state "1910-1930 was the lowest minimum in the 164 year record."

*Response: Thanks for your suggestion. We revised it.*

Line 24 Should be "The change in trend"

*Response: Thank you for the suggestion. We corrected it.*

Line 43 Should this be "Ocean heat content" …and dynamic processes.

*Response: Thank you for the suggestion. We corrected it.*

Line 59 add a comma after " droughts" and remove " and"

*Response: Thank you for the suggestion. We corrected it.*

Line 92 Replace "space " by "research"

*Response: Thank you for the suggestion. We corrected it.*

Line 116 Replace " are " by "is"

*Response: Thank you for the suggestion. We corrected it.*

Line 153 Replace "in the north" by " to the north"

*Response: Thank you for the suggestion. We corrected it.*

Line 189 Replace "Perform a significance test" by "A significance test is performed…"

*Response: Thank you for the suggestion. We corrected it.*

Line 210 Figure 3 (top graph) I was surprised that the domain covers 0-60N with temperatures ranging from 3-6C in the north to 26 to 28 C but the mean is about 26 C ? Need to check this is correct.

*Response: Thank you for the suggestion. We corrected it.*

Line 213 Legend " All the trends are significant" not " is siginificant.

*Response: Thank you for the suggestion. We corrected it.*

Line 218 Should be North Western Pacific"

*Response: Thank you for the suggestion. We corrected it.*

Line 225 Should be " 95% significance test "

*Response: Thank you for the suggestion. We corrected it.*

Line 238 Should be " red lines are their trends"

*Response: Thank you for the suggestion. We corrected it.*

Line 239 -240 I suggest removing "The same as the annual pattern, seasonal pattern" replacing by " The seasonal pattern for the latest 30 years shows a more significant warming trend than that over the 164 year period."

*Response: Thank you for the suggestion. We corrected it.*

Line 243 Insert " is" after anomaly.

*Response: Thank you for the suggestion. We corrected it.*

Line 251 Delete " curve"

*Response: Thank you for the suggestion. We corrected it.*

[revised manuscript text omitted]